# Comparison of Atmospheric and Lithospheric Culturable Bacterial Communities from Two Dissimilar Active Volcanic Sites, Surtsey Island and Fimmvörðuháls Mountain in Iceland

**DOI:** 10.3390/microorganisms11030665

**Published:** 2023-03-06

**Authors:** Aurélien Daussin, Pauline Vannier, Marine Ménager, Lola Daboussy, Tina Šantl-Temkiv, Charles Cockell, Viggó Þór Marteinsson

**Affiliations:** 1Faculty of Food Science and Nutrition, University of Iceland, 113 Reykjavík, Iceland; 2MATIS, Department of Research and Innovation, 113 Reykjavík, Iceland; 3University of Technology of Compiègne, 60203 Compiègne, France; 4Department of Biology, Aarhus University, 8000 Aarhus, Denmark; 5Arctic Research Center, Department of Biology, Aarhus University, 8000 Aarhus, Denmark; 6iCLIMATE Aarhus University Interdisciplinary Centre for Climate Change, Department of Environmental Science, Aarhus University, 8000 Aarhus, Denmark; 7School of Physics and Astronomy, University of Edinburgh, EH8 9YL Edinburgh, Scotland; 8Agricultural University of Iceland, Hvanneyri, 311 Borgarnes, Iceland

**Keywords:** bioaerosols, microbial cultivation, atmospheric bacteria, volcanic rocks, survival, colonization, source apportionment, air-mass trajectory

## Abstract

Surface microbes are aerosolized into the atmosphere by wind and events such as dust storms and volcanic eruptions. Before they reach their deposition site, they experience stressful atmospheric conditions which preclude the successful dispersal of a large fraction of cells. In this study, our objectives were to assess and compare the atmospheric and lithospheric bacterial cultivable diversity of two geographically different Icelandic volcanic sites: the island Surtsey and the Fimmvörðuháls mountain, to predict the origin of the culturable microbes from these sites, and to select airborne candidates for further investigation. Using a combination of MALDI Biotyper analysis and partial 16S rRNA gene sequencing, a total of 1162 strains were identified, belonging to 72 species affiliated to 40 genera with potentially 26 new species. The most prevalent phyla identified were Proteobacteria and Actinobacteria. Statistical analysis showed significant differences between atmospheric and lithospheric microbial communities, with distinct communities in Surtsey’s air. By combining the air mass back trajectories and the analysis of the closest representative species of our isolates, we concluded that 85% of our isolates came from the surrounding environments and only 15% from long distances. The taxonomic proportions of the isolates were reflected by the site’s nature and location.

## 1. Introduction

Extreme meteorological events such as volcanic eruptions, dust storms, and wildland fires can aerosolize a large number of viable microbes up to tens of kilometres high in the atmosphere that can be transported short or long distances before landing back on the Earth’s surface [1,2,3,4]. Bioaerosols that reach the free troposphere are typically subject to long-range dispersal up to thousands of kilometres [5], whereas those that remain in the lower and most vertically mixed part of the troposphere called the planetary boundary layer account for most local dispersal [6]. During their dispersal through the atmosphere, cells experience stressful conditions including solar radiation, oxidative stress, osmotic shock, and freeze–thaw cycles [7]. Those that survive and remain active under such extreme conditions must possess strong capacities to withstand stress such as cold adaptation and freeze–thaw cycles resistance. For instance, microbial pigment production has been shown to provide a selective advantage against UV radiation [8,9]. These natural pigments can be used, for example, to develop novel drug compounds as well as in the textile and food industries [10]. Airborne microorganisms not only have a biotechnological interest, but they might also play part in atmospheric processes such as cloud and precipitation formation, have a negative impact on human health and agriculture [11], and may be included in meteorological and climatic models [12,13,14]. When these airborne microorganisms finally deposit on terrestrial environments via sedimentation (dry deposition) or precipitation (wet deposition) [15], they must face autochthonous microbial communities. Pigment production seems to not only protect from UV radiation during atmospheric transportation but possesses biological properties such as antimicrobial and antiviral activities [16,17,18] that give airborne microbes an advantage over these already settled surface communities.

With its active volcanoes and freshly formed lava field, Iceland is a perfect study site for the investigation of microbial establishment and succession on newly formed volcanic rocks. One of the first studies of microbial colonization on freshly formed volcanic rocks focused on Surtsey island, formed during the 1963–1968 volcanic eruptions off the southern Icelandic coast [19]. Phototrophs were already observed by 1968 and subsequent culture-based and microscopy investigations reiterated the importance of chlorophytes, lichens, and mosses to ecosystem development on the island. A further study reported the presence of cyanobacteria, including Anabaena and Nostoc on the Icelandic island of Heimaey, 18 months after a volcanic eruption in 1973 [20]. A recent study showed that microbes found on the surface of Surtsey come from diverse sources including the deep subsurface, surrounding seawater, and other surface ecosystems [21]. A combination of molecular and culture-based methods has been used to study succession on the Fimmvörðuháls lava field that formed in 2010 after the eruption of Eyjafjallajökull. The study observed that early microbial colonists were not composed primarily of phototrophs but rather of microorganisms, such as Betaproteobacteria, affiliated with known diazotrophs, chemolithotrophs and heterotrophs [20]. Though the air is a major source for the dispersal of microbial cells that establish in newly emerged volcanic environments, the processes of dispersal and succession in volcanic settings have so far not been studied in parallel.

The objectives of this study were (i) to assess the culturable microbial diversity of air and lava rock samples of two Icelandic volcanic sites for the first time, Surtsey island and the inland lava flow Fimmvörðuháls; (ii) to investigate or predict the origin of the isolates; and (iii) to select candidates that have the potential to be dispersed through the air. This selection process will be based on known traits that are likely to enhance the success of colonization after aeolian dispersal. By predicting the origin of the isolates, we can identify their potential sources. This information can improve our understanding of the distribution and abundance of these microorganisms and their role in various ecological processes. For instance, it will help us identify and mitigate any negative impacts they may have on ecosystems and their inhabitants. Moreover, investigating the microbial diversity of Icelandic volcanic sites can provide insight into the survival and dispersal of microorganisms in extreme environments and have implications for understanding the spread of microorganisms in other environments.

More than 1000 isolated microbial strains from air and lava rock samples were isolated and identified using genetic analysis. Air-mass back trajectories prediction and literature-based data were used to draw hypotheses about these isolates’ sources and survival properties.

## 2. Materials and Methods

### 2.1. Site Description and Sample Collection

Air and lava rock samples were collected in 2018 and 2019 from two different Icelandic volcanic sites: Surtsey (rock samples: 19–22 July 2018 and air and rock samples: 18–22 July 2019) and Fimmvörðuháls (air samples: 12–14 September 2018 and rock samples: 27–28 September 2019) (Figure 1). Surtsey (63°18′11″ N, 20°36′11″ W) is a neo-volcanic island on the south coast of Iceland which was formed after a four-year submarine volcanic eruption from 1963 to 1967. The recent Fimmvörðuháls lava is at 1100 m altitude (63°37′53″ N, 19°26′50″ W) and located between the Eyjafjallajökull and Mýrdalsjökull glaciers in southern Iceland. This lava was produced during the eruption of the Eyjafjallajökull volcano in the spring of 2010. Each of these sampling sites was assessed using several sampling stations: five stations on Surtsey and three stations at Fimmvörðuháls. The coordinates and characteristics of the stations can be found in Table 1.

Air samples were collected in duplicates using high-flowrate (Kӓrcher DS6 Waterfilter) impingers (Appendix A). The sampler was cleaned with ethanol prior to the experiment whereas the vortex chambers, where the airborne microbial cells are trapped in the sampling liquid, were autoclaved. The air was vacuumed into 2.5 L of 1× phosphate-buffered saline (PBS) solution for 5 h at around 3100 L/min [22]. For each run, 500 mL of sampling solution was collected as a negative control after 5 min of sampling. The purpose of running the instrument for 5 min is to induce the vortex of the sampling liquid, which allows the liquid to come into contact with all surfaces within the vortex chamber prior to being sampled as a negative control. We can therefore thoroughly address the background contamination in the sampler and the sampling liquid [22]. After 5 h, part of the sampling solution was aliquoted into 50 mL tubes for cultivation purposes and stored at 4 °C in the dark. In 2018, basaltic lava rocks from each location and site were chosen and autoclaved on-site using a portable autoclave. The sterilized rocks were placed back at their initial location and marked using vertical metallic sticks which were placed in the ground. To assess the colonisation after one-year exposure, these 16 sterilized rocks were sampled after one year and stored at 4 °C in the dark prior to analysis and are hereof designated as “one-year-old lava rocks”. To compare these colonizers communities with autochthonous communities, two non-autoclaved nine-year-old basaltic rocks were collected from the environment at sites 1 and 3 of Fimmvörðuháls in 2019 and are hereof designated as “nine-year-old lava rocks”. The information on samples collected at different stations per year is described in Table 1.

### 2.2. Isolation, Characterization and Grouping by MALDI-TOF Mass Spectrometry of the Microbial Strains

Samples were initially inoculated on solid Reasoner’s 2A (R2A) agar (BD DifcoTM) and Blue Green Algae (BG11) agar (BD DifcoTM), at 4 °C, 10 °C (close to the Icelandic air temperature) and 22 °C. The first analyses revealed that the most diverse communities were growing at 22 °C in R2A and we kept these parameters for the study as we needed to implement limitations. R2A medium has been used to cultivate microorganisms isolated from diverse natural environments (air [23,24], surface soils of the Surtsey island [25]). For air samples, 100 µL of the 1xPBS solution and the negative control from the impinger were inoculated on solid medium plates and enrichment cultures were incubated at 22 °C until growth occurred. Additionally, solid R2A plates without inoculation were used as negative controls of the medium and the experiments. According to a modified version of the protocol from Kelly et al. [20], the lava rock samples were crushed to powder under sterile conditions using a cold sterile mortar and pestle. Briefly, 5 g of rock powder was diluted in 50 mL of 1X PBS solution for 24 h under 100 rpm agitation at 4 °C. A volume of 100 µL of this PBS solution was then used as an inoculum on solid medium plates. Plates were incubated at 22 °C until growth occurred. After growth, visually different colonies were selected and purified with repetitive streaking. Pure strains were stored in 20% glycerol solution at −140 °C at the Icelandic Strains Collection and Records (ISCAR, http://iscar.matis.is/ accessed on 10 April 2022)

All the isolated strains were screened using a Microflex™ MALDI-TOF mass spectrometer (Bruker, Billerica, MA, USA). A mass spectrum (MSP) dendrogram was created using MALDI Biotyper 3.1 software (Bruker, Billerica, MA, USA) based on m/z spectra comparison. Strains with a distance level above 400 were considered a unique taxonomic group whereas strains that showed a distance level below 400 were considered the same strain or species [26]. To optimize the experiments, for each unique taxonomic group, the fastest-growing isolate was chosen and considered to be the representative strain for the group itself. All further analyses were done on the representative strains.

### 2.3. Identification of Group Representatives by 16S rRNA Gene Sequencing

The DNA of the representative strains was extracted using a 6% Chelex 100 Solution (Biorad, Hercules, CA, USA) according to the manual of the manufacturer. The V1–V4 region of the bacterial 16S rRNA gene was amplified from each DNA using the F9 (“5-GAGTTTGATCCTGGCTAG-3”) and R805 (“5-GACTACCCGGGTATCTAATCC-3”) universal primers [27] and the OneTaq™ DNA polymerase (New England Biolabs, Ipswich, USA). The V4 region of the eukaryotic 18S rRNA gene was amplified using the 1F (“5-AACCTGGTTGATCCTGCCAGT-3”) and 1528R (“5-TGATCCTTCTGCAGGTTCACCTAC-3”) universal primers (Medlin et al. (1988) Genetica 71: 491–499.) and the OneTaq™ DNA polymerase (New England Biolabs, Ipswich, MA, USA). Both amplification reactions were carried out in a final volume of 25 µL with 1X One Taq GC reaction Buffer, 1X One Taq high GC Reaction Enhancer, 0.2 µM dNTPs, 0.4 µM of each primer, 0.2 mM of MgCl2, 0.625 U/µL of OneTaq Hot Start DNA Polymerase and 2 ng/µL of DNA template. PCR was run on a ProFlex™ Thermal cycler (Applied Biosystems™, Foster City, CA, USA). Respective PCR conditions were as follows for the prokaryotic or the eukaryotic DNA: an initial denaturation step at 94 °C for 1 min or 5 min, followed by 35 cycles of denaturation at 94 °C for 40 s or 2 min, annealing at 52 °C for 40 s or 50 °C for 2 min, and extension at 72 °C for 1 min or 4 min, with a final extension step of 7 min at 72 °C. PCR products were verified by electrophoresis on a 1% agarose gel stained with SYBR safe (Thermo Fisher Scientific, Waltham, MA, USA) for 40 min at 100 V in 1 × Tris Acetate EDTA (TAE) buffer. The PCR products were cleaned for sequencing through the Exo-SAP™ Purification (New England Biolabs, Ipswich, USA) and the BigDye™ Terminator v3.1 (Applied Biosystems™, Foster City, CA, USA) processes. Sanger sequencing was then performed on the cleaned amplified 16S rRNA genes using the 3730 DNA Analyser (Applied Biosystems™, Foster City, CA, USA). The sequences were processed using the Sequencher^®^ version 5.4.6 DNA sequence analysis software (Gene Codes Corporation, Ann Arbor, MI, USA). All reads were trimmed to obtain at least 200 bp length sequences and 90% quality. To determine their taxonomic affiliation, the trimmed sequences were compared with the NCBI 16S and 18S ribosomal RNA sequences databases and the used NCBI BLASTN alignment (https://blast.ncbi.ncbi.nlm.nih.gov/Blast.cgi, accessed on 20 March 2022) that was optimized for highly similar sequences.

### 2.4. Construction of Phylogenetic Trees

One representative 16S rRNA gene sequence of each taxonomic group was selected to build a phylogenetic tree with the type species of 40 different genera. Sequences were classified and aligned using the online portal of the SILVA Incremental Aligner (SINA 1. 2. 11) tool of the ARB-Silva database (http://www.arb-silva.de/aligner/, accessed on 20 August 2022) [28]. The tree computed with RAxML was imported in Interactive Tree Of Life (iTOL) v6.4 [29].

Representative sequences of the novel strains (<98.65% 16S rRNA gene sequence similarity) isolated in this study were selected and aligned against 16S rRNA gene sequences of the closest cultured type strains using the MUSCLE algorithm. The phylogenetic tree was built using the maximum-likelihood method with a Kimura 2-parameter model and 1000 bootstrap replications, using MEGAX.

### 2.5. Comparison of the Microbial Communities Using Statistical Analyses

A non-metric multidimensional scale (NMDS) plot was made to visualize the distance between the microbial communities of our samples in a multidimensional space, using the readxl [30], vegan [31], ggplot2 [32], ggforce [33] and ggrepel [34] packages. A Pearson correlation matrix was built to confirm the correlations between the different communities using the readxl [30], reshape2 [35], ggplot2 [32], and corrplot [36] packages. Finally, the dissimilarities between the microbial communities of the different types of samples, the sites, the stations, and the sampling years were calculated using the ANOSIM statistic R and the significance using the vegan [31] package. The Bray–Curtis distance was used to calculate dissimilarity and the number of permutations was set to 9999. All statistical analyses were performed on RStudio using R (v 3.6.0).

### 2.6. Predictive Sources of the Cultivated Isolates

Predictive sources of the cultivated isolates on both sites were determined by combining the air mass back trajectories and the analysis of the closest representative species of our isolates based upon the literature.

In the context of this study, the distance for the surrounding environments corresponds to a maximum of about 10 km around the sampling location whereas long distance corresponds to a minimum of about 10 km around the sampling location. This distance was selected arbitrarily as a representative boundary within which the surrounding environment does not change significantly. For example, the next island is more than 10 km away from Surtsey and within a 10 km radius around Fimmvörðuháls, the environment is primarily composed of glaciers and natural terrain whereas, beyond this boundary, the landscape shifts to include roads and villages. Air mass trajectory calculations were performed to establish the likely long-range source regions of the microbes identified from air and rock samples using the Hybrid Single-Particle Lagrangian Integrated Trajectory model (HYSPLIT) and the GDAS one-degree archives for Iceland [37,38]. A two-day backward trajectory was calculated for air parcels arriving at three different heights, each representing a different air layer. For Fimmvörðuháls: 1000 m (sampling height, ground influence), 1500 m (the upper part within the boundary layer), and 2000 m (above the boundary layer). For Surtsey: 50 m (sampling height, ground influence), 500 m (the upper part within the boundary layer), and 1500 m (above the boundary layer). Assuming possible mixing events between different air layers, the air mass sampled on the ground might represent a mixing of the air layers, and therefore the origins of the isolated microbes might be related to the origins of the different air masses [39]. The environment of origin was determined for the closest strains of the 72 cultivable species (>98.5% BLAST identity) recording the habitat of their closest relatives as reported in the literature. To assess the local sources of the isolates, our approach was to analyse the characteristics of the environment within 10 km of the sampling site by looking at factors such as the type of terrain, the presence of human settlements, and the types of flora and fauna present.

### 2.7. Physiological Survival Properties of the Isolates

Information about the closest representatives of the 72 identified species were gathered by reviewing the literature, with a focus on assessing their ability to withstand various environmental stressors and their potential to be deposited among established surface communities. Properties assessed included pigment production, cold adaptation capacity, radiation and freeze–thaw cycle tolerance, and antimicrobial production capacity.

## 3. Results

### 3.1. Isolation, Characterization and Grouping by MALDI-TOF Mass Spectrometry of the Microbial Stains

From the 38 environmental samples in the two studied locations, Surtsey and Fimmvörðuháls, a total of 1162 strains were isolated. Different colony morphologies were chosen for preliminary identification using the MALDI Biotyper. No colony was observed on the negative control plates for the air samples.

After grouping the isolated strains with a distance level under 400 on the MSP dendrograms, 269 different strains were selected and identified by using 16S rRNA gene sequencing. On average, a MALDI group represents five isolated strains. The representative strains of each MALDI group were selected for partial 16S rRNA gene sequencing resulting in 72 different species affiliated to 40 genera with potentially 26 new species (Figure 2 and Figure 3, Appendix A).

### 3.2. Phylogenetic Analysis of the Cultivated Microbial Isolates

All samples, air and one-year-old and nine-year-old lava rocks, harboured cultivable strains belonging to seven different phyla with Proteobacteria and Actinobacteria found in all samples (Figure 4).

Among the 40 genera identified in this study, 10 were found in common between the island of Surtsey and the highland Fimmvörðuháls, and seven were found in common between the air and the one-year-old lava rock samples. A total of six genera were found in common between both the sites and the type of samples: *Pseudomonas*, *Paracoccus*, *Pedobacter*, *Bacillus*, *Micrococcus* and *Moraxella* (Figure 5A).

Additionally, bacteria belonging to six different genera were cultivated from both one-year-old and nine-year-old lava rocks at Fimmvörðuháls (Figure 5B).

### 3.3. Comparison of the Microbial Communities Using Statistical Analyses

The NMDS ordination plot analysis showed that communities found in the air and the one-year-old lava rocks as well as in Surtsey and Fimmvörðuháls are distant (Figure 6A). A significant dissimilarity between these rocks and the air communities was confirmed with the ANOSIM statistical test, whereas a lower dissimilarity and no significance was found between the island and the mountain sites communities. The Pearson correlation matrix, which is presented in Figure 6B, supports this result with the highest correlation between communities in the one-year-old lava rock samples from two different sites. Moreover, the air of Surtsey seems to host its own cultivable microbial community with a low Pearson correlation to any other environment (Figure 6B, blue colour).

### 3.4. Predictive Sources of the Cultivated Isolates

The air–mass trajectories 48 h before the samplings at both sites came from the west in 2018 and from the east in 2019. In both cases, air mass travelled above most marine environments; however, for the 2019 sampling for the air mass reaching Fimmvörðuháls, there was a small terrestrial contribution from its time passing over the southern part of Iceland before it reached Scandinavia and the sampling sites (Appendix A).

Surtsey is an island surrounded by seawater and exposed soil and plants are present in the area. Therefore, seawater, plant, soil, air and humans were considered local sources. Based on the analysis of the closest representative species of our isolates from Surtsey in the literature, we postulate that 89% of them came from local sources (40 strains of the 45 cultivated ones) (Figure 7A) with almost half of it previously isolated from soil, as e.g., *Lysinimonas soli* [40], and one third previously isolated from seawater, as e.g., *Paracoccus aquimaris* [41] and *Flavobacterium frigidimaris* [42]. Only five isolates were previously isolated from sources that are not present locally. For example, *Sphingomonas glacialis* was previously isolated from a glacier [43].

Fimmvörðuháls is a path between glaciers, with a famous hiking track passing nearby, and exposed soil and plants are present in the area. Glaciers, air, plants, soil, and humans were considered local sources. Based on the analysis of the closest representative species of our isolates from Fimmvörðuháls in the literature, we postulate that 86% of them came from local sources (18 of the 21 cultivated strains) (Figure 7B). The strains with soils and humans as potential sources were dominant (Respectively, 9 and 7 out of 18) such as *Streptomyces candidus* (soil, [44]) and *Staphylococcus epidermidis* (human, [45]). *Noviherbaspirillum psychrotolerans* has been previously isolated from a glacier [46]. Only three isolates have been previously isolated from sources that are not present locally. For example, *Paracoccus marinus* was previously isolated from seawater [47].

### 3.5. Physiological Survival Properties of the Isolates

Overall, 53% of the isolated strains possess atmospheric transportation survival properties. For instance, *Pseudomonas turukhanskensis* produces beige pigments, *Micrococcus flavus* yellow pigments, *Sporobolomyces blumeae* orange pigments, *Janthinobacterium rivuli* purple pigments, and *Pedobacter roseus* pink-colour pigments (Table 2). A total of nine (13%) of the isolate’s representatives are known to be cold adapted, such as *Flavobacterium frigidimaris* and *Noviherbaspirillum psychrotolerans* (Table 2). A smaller proportion of the isolates produces antibiotics (*Streptomyces candidus*) or antifungal compounds (*Stenotrophomonas rhizophila*) and have shown resistance against atmospheric stress such as radiation (*Deinococcus taklimakanensis*), dry conditions (*Pseudarthrobacter siccitolerans*), and freeze–thaw cycles (*Sporobolomyces blumeae*) (Table 2).

## 4. Discussion

Our study has revealed new insights into the diversity and origin of culturable microorganisms in the air and on lava rocks in Iceland. By comparing the microbial communities from air, from one-year-old rocks, and from nine-year-old rocks from two different volcanic regions (Fimmvörðuháls and Surtsey), we have identified some of the factors that may influence the assemblage of these communities.

One of the main characteristics of the microbial communities that we identified is the dominance of certain phyla, including Proteobacteria, Actinobacteria, Firmicutes, and Bacteroides. These phyla are known to be common in Icelandic rocks and air samples [20,21,83,84,85,86,87,88], and their presence in our samples is consistent with previous studies. *Deinococcota* representatives were only isolated from one-year-old lava rock samples in our study which agrees with other studies [89]. However, representatives of this phylum are also usually acquired from the air [90,91] but no strain was isolated from our air samples. In addition to these phyla, we also observed cultivated fungi in the air samples, including *Ascomycota* and *Basidiomycota* strains. During the dissolution of the rocks into PBS prior to inoculation, it is possible that toxic compounds were released from the rocks, which may have played a role in the absence of fungal recovery from these samples [92]. Moreover, these fungi have been shown to dominate atmospheric communities in central Europe and Antarctica [93,94], and their presence in our air samples suggests that fungi may also play a significant role in shaping the microbial communities in Icelandic air.

At the genus level, we observed significant differences between the air and rock samples, which we attribute to the different time scales at which these microbial communities assemble. The air is a very dynamic system where microbial cells from diverse sources are mixed on short time scales [2,95], while lava rocks contain microbial communities that assemble over years and decades. However, we also observed some similarities in the diversity of both one-year-old and nine-year-old lava rocks in Fimmvörðuháls, suggesting that airborne communities can develop on the lava rocks after one year of exposure. This indicates that airborne microorganisms may play a role in shaping the microbial communities on lava rocks over time, though further research is needed to confirm this. Moreover, both rocks are touching the ground and it is possible that the microorganisms isolated from them originated from the soil, which could account for the shared genera found on the one- and nine-year-old rocks. The environmental conditions in the air and on the lava rocks are comparable to some extent: the microbiome must survive diverse stresses such as desiccation and re-hydration, temperature fluctuations, lack of nutrients, and UV radiation. These are common stressors associated with surface–atmospheric locations and they could explain why most of the isolates in this study originated from soil, plant and seawater.

When we compared the microbial communities from the different regions and sample types, we found that most microbes came from local sources. For example, the air of Surtsey was found to host its own unique microbial communities with a low Pearson correlation to the other studied environments, which were dominated by species from short-distance sources such as the surrounding seawater. This is likely due to the remote location of Surtsey and its proximity to the sea, which may contribute to the singularity of its marine air communities. Additionally, seawater-related species such as *Flavobacterium frigidimaris* [42] and *Paracoccus aquimaris* [41] were recovered from one-year-old lava rocks in Surtsey and could have reached the rock samples through the subsurface where, for instance, other *Flavobacteriales* were isolated [21].

One of the questions that our study sought to address was the extent to which microbes can be transported from long-distance sources and survive in new environments. Our results suggest that only a small proportion of the isolates in our study were transported from long-distance sources. This may be due to the difficulty of microbes to survive long-range dispersal, as many factors can affect their survival and ability to colonize new environments [7,28,96]. However, our results also suggest that some microbes are able to survive long-range transport and establish themselves in new environments, such as the seawater-related species *Paracoccus marinus* [47] and *Pseudomonas prosekii* [97], which were recovered from Fimmvörðuháls despite being relatively far from the sea.

As the adaptation of bacteria to new environments is generally slower than the rate at which their environments can change, these stress-resistant species might already have some adaptations to environmental extremes and may be better able to withstand changes in their habitat than those that do not. This is because they may be more resistant to stress or more able to utilize different resources, which can give them a competitive advantage in new environments [98]. A metatranscriptomic study has highlighted that the physiological response of airborne microorganisms to stress is mainly linked to membrane modifications, iron uptake, and synthesis of cryo- and osmoprotectant [99]. Further research will be needed to fully understand the mechanisms by which airborne microbes are able to survive and colonize new environments, and to identify the factors that influence their success or failure.

## 5. Conclusions

In this study, we assessed and compared for the first time the culturable microbial communities from two dissimilar active volcanic sites in Iceland: Surtsey and Fimmvörðuháls. Overall, our results suggest that the microbial communities in the air and on lava rocks are shaped by a combination of factors, including the time scales at which they assemble, the environmental conditions they are exposed to, and the sources of microbes that contribute to their formation. In this study, we have only touched the tip of the iceberg because we have only investigated the culturable communities and not the whole microbiome using cultivation-independent approaches. Thus, further research will be needed to fully understand the role of airborne microorganisms in shaping these communities, and in order to identify the mechanisms by which these microorganisms are able to survive and colonize new environments.

The use of R2A medium for cultivating the microbes is one of the limitations of this study. While this medium is widely used and suitable for many types of bacteria, it may not support the growth of all microbes present in the environments sampled. Additionally, the samples were not always taken simultaneously, which means that some of the microbial differences observed may be due to temporal changes rather than the specific environment. Another limitation is the versatility of the air mass trajectories, which can affect the microbial composition of the environments. The changing weather conditions during the study time frame may also have had an impact on the microbial communities studied. Finally, the origins of the microorganisms are based on the assumption of short-range and long-range dispersal, however, it is acknowledged that the ocean, wind, and humans can transport microorganisms over vast distances. Therefore, it is possible that the microorganisms found in this study may have originated from a location farther away than what was assumed based on the surrounding environment. Overall, these limitations highlight the need for caution in interpreting the results of this study and underscore the importance of considering the potential influences of multiple factors on microbial communities.

However, based on the sources and the physiological survival properties of the cultivated isolates presented in this study, a list of candidates that are likely to encounter aerosolization/deposition processes was made: *Streptomyces candidus*, *Paracoccus marinus*, *Janthinobacterium rivuli*, *Stenotrophomonas rhizophila*, *Pedobacter nototheniae*, *Deinococcus taklimakanensis*, *Paenibacillus liaoningensis*, *Sporobolomyces blumeae*, and *Saraclodium kiliense*. These candidates are available in culture and can therefore be investigated in detail for their ability to survive atmospheric stress factors which might further expand the value of this study.

## Figures and Tables

**Figure 1 microorganisms-11-00665-f001:**
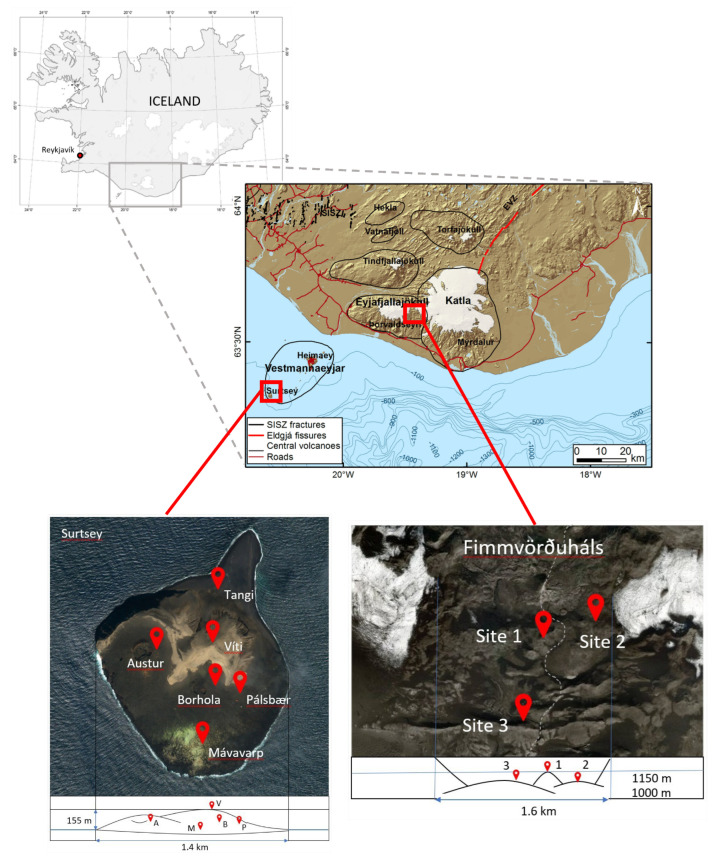
Locations of the Surtsey and Fimmvörðuháls sampling sites and the stations.

**Figure 2 microorganisms-11-00665-f002:**
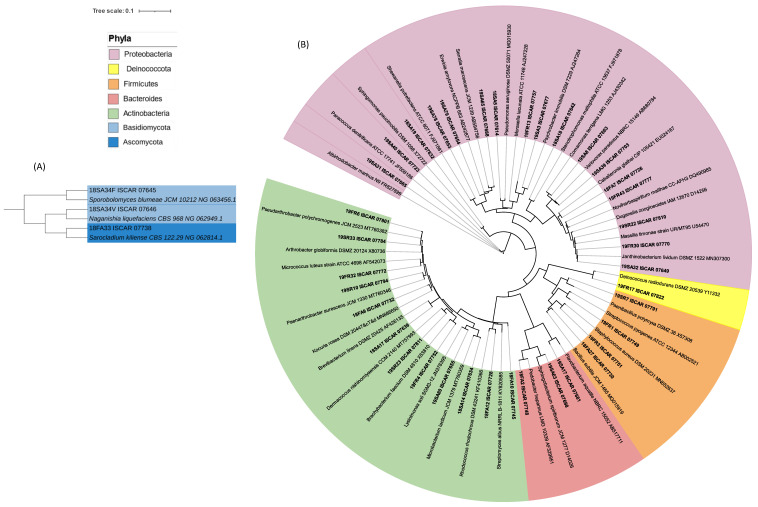
Phylogenetic tree of the partial 16S rRNA gene sequences of 40 strains cultivated in this study and type species of 40 different genera. (**A**) Eukaryotic strains. (**B**) Prokaryotic strains. One representative 16S rRNA gene sequence of each taxonomic group was selected to build a phylogenetic tree. Sequences were classified and aligned using the online portal of the SILVA Incremental Aligner (SINA 1. 2. 11) tool of the ARB-Silva database (http://www.arb-silva.de/aligner/, accessed on 20 August 2022). The tree computed with RAxML was imported in Interactive Tree of Life (iTOL) v6.4.

**Figure 3 microorganisms-11-00665-f003:**
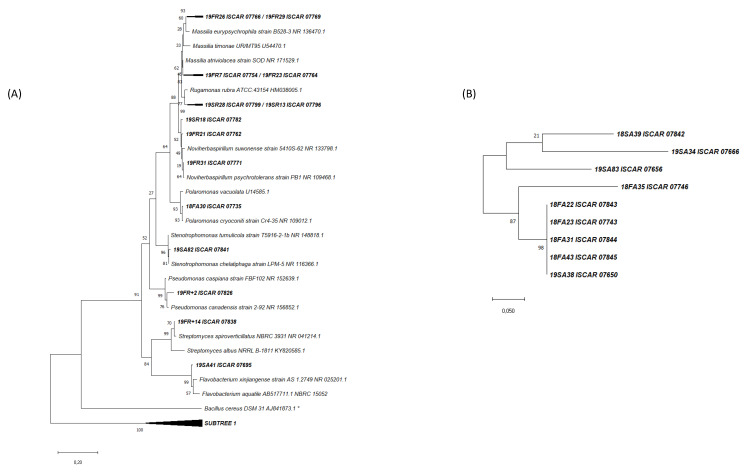
(**A**) Phylogenetic tree based on partial 16S rRNA gene sequences showing the relationship between the novel species isolated in this study and closest cultured strains. Sequences were aligned using MUSCLE algorithm and the tree was built using the maximum-likelihood method with a Kimura 2-parameter model and 1000 bootstrap replications, using MEGAX. A total of 20 sequences from this study were aligned with 17 from the literature. * Outgroup (*Bacillus cereus*). (**B**) Subtree 1 showing the relationship between the non-affiliated novel strains isolated in this study.

**Figure 4 microorganisms-11-00665-f004:**
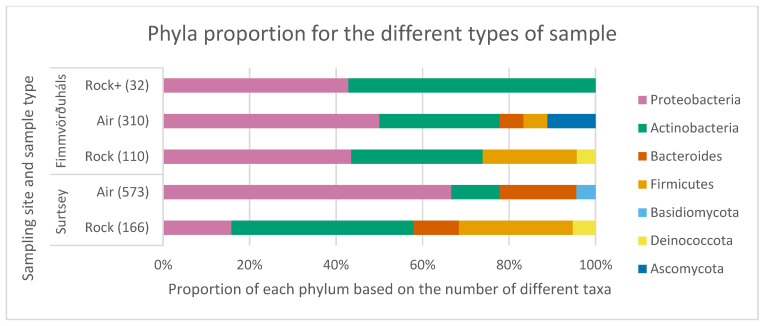
Stacked bar plot showing the proportion of the different phyla in each sampling site and type. The numbers under brackets represent the number of isolates. The eukaryotes phyla appear in blue. Rock represents the one-year-old lava rocks whereas Rock+ represents the nine-year-old lava rocks. Adapted for colour blindness.Cultivable representatives of the phylum Bacteroides were found in all samples except the one-year-old lava rocks of Fimmvörðuháls, and *Firmicutes* in all samples except the air of Surtsey. Members of Deinococcota were found in the one-year-old lava rock samples at both sites but not in the air and the nine-year-old lava rock of Fimmvörðuháls. Eukaryotic phyla were only found in the air with a member of *Ascomycota* cultivated from the air of Fimmvörðuháls and a member of *Basidiomycota* cultivated from the air of Surtsey.

**Figure 5 microorganisms-11-00665-f005:**
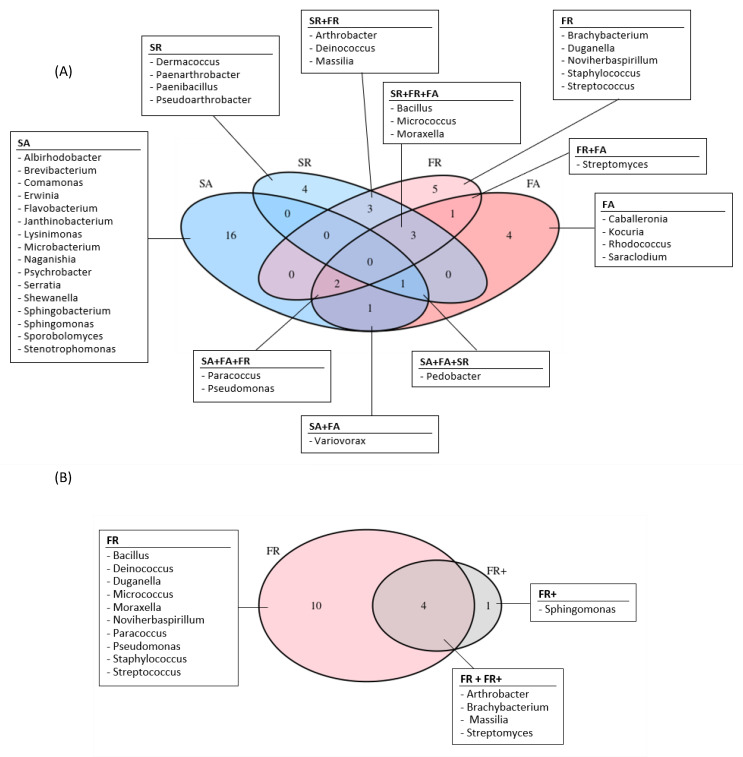
Venn Diagrams of the genera detected by cultivation. (**A**) Air and one-year-old lava rock samples. (**B**) One- and nine-year-old lava rocks from Fimmvörðuháls. S: Surtsey, F: Fimmvörðuháls, A: air, R: one-year-old lava rocks, R+: nine-year-old lava rocks.

**Figure 6 microorganisms-11-00665-f006:**
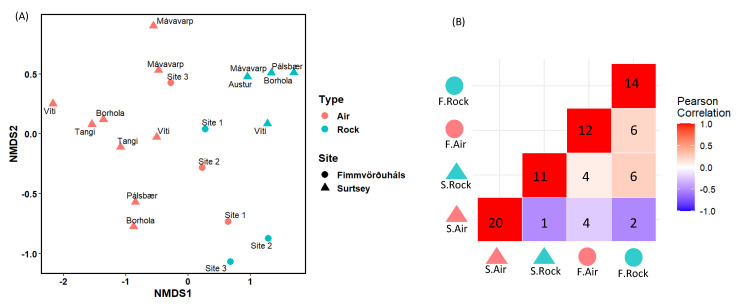
(**A**) NMDS ordination of the cultivable microbial community composition in the different stations of the sampling sites, based on the different genera cultivated from the air samples and the one-year-old lava rock samples. The ANOSIM statistic R and the significance for the type of sample, the sites, the stations, and the years were 0.5851/0.0001, 0.1063/0.1104, −0.1713/0.8798, and 0.07837/0.1496, respectively. (**B**) Pearson correlation matrix based on the cultivated genera for each sampling. Numbers represent the cultivated genera in the one-year-old lava rocks of Surtsey (S. Rock), the one-year-old lava rocks of Fimmvörðuháls (F. Rock), the air of Surtsey (S. Air) the air of Fimmvörðuháls (F. Air), and those in common between these samples. Statistical analyses were performed on RStudio using R (v 3.6.0).

**Figure 7 microorganisms-11-00665-f007:**
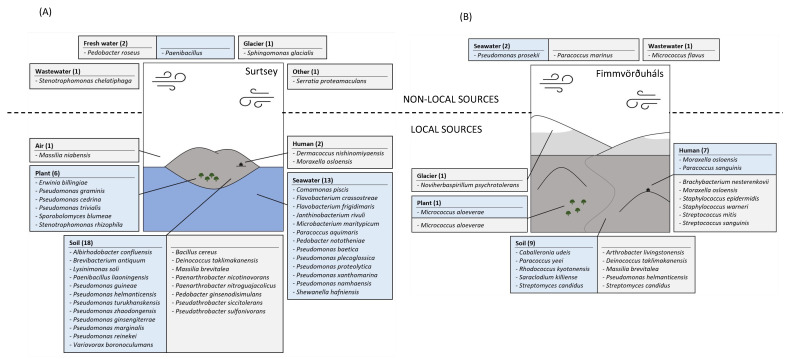
Overview of the predictive sources of the species-determined taxa cultivated from the air (blue) and the one-year-old lava rocks (grey) of Surtsey (**A**) and Fimmvörðuháls (**B**). The numbers represent the count of the different taxa per source.

**Table 1 microorganisms-11-00665-t001:** Characteristics of the sampling stations and number of samples.

Location	Station-Abbreviation	Coordinates	Elevation	Characteristics	Year	Air Samples	Rock Samples	
							1-Year-Old	9-Year-Old
Surtsey	Mávavarp-M	63°17.880′ N 20°36.092′ W	50 m	Gull nesting area-vegetation	2018	2	0	0
					2019	2	2	0
	Borhola-B	63°18.097′ N 20°36.010′ W	66 m	Boreholes-Human activity	2018	2	0	0
					2019	2	2	0
	Viti-V	63°18.244′ N 20°36.062′ W	166 m	Lighthouse–Highest point	2018	2	0	0
					2019	2	2	0
	Tangi-T	63°18.455′ N 20°36.011′ W	11 m	Peninsula–Rocks	2018	2	0	0
					2019	2	0	0
	Pálsbær-P	63°18.047′ N 20°35.828′ W	38 m	Human hut	2018	0	0	0
					2019	2	2	0
	Austur-A	63°18.153′ N 20°36.609′ W	108 m	Crater	2018	0	0	0
					2019	0	2	0
Fimmvörðuháls	Top Eyjafja-1	63°37.985′ N 19°26.466′ W	1036 m	Crater	2018	2	0	0
					2019	0	2	1
	Lava Eyja-2	63°38.182′ N 19°25.991′ W	1030 m	Lava field	2018	2	0	0
					2019	0	2	0
	Hut-3	63°36.654′ N 19°26.489′ W	875 m	Human hut	2018	2	0	0
					2019	0	2	1

**Table 2 microorganisms-11-00665-t002:** Survival properties of the identified species based upon the literature. Yellow colour: possess survival properties. NCBI 16S ribosomal RNA sequences database was used on BLAST. * Previously isolated from human.

Strain Name	Closest Match on BLAST		Survival Properties			
	Genus	Species	Pigment Production	Cold-Adaptation	Other Properties	Reference
19SA31	*Albirhodobacter*	*confluentis*	-	*-*	*-*	-
19FR6	*Arthrobacter*	*livingstonensis*	-	+	*-*	[48]
19FR + 13	*Arthrobacter*	*koreensis*	+, yellow	*-*	*-*	[49]
18FA27	*Bacillus*	*cereus*	-	*-*	*-*	-
19FR + 12	*Brachybacterium*	*rhamnosum*	-	*-*	*-*	-
19FR4	*Brachybacterium*	*nesterenkovii*	+, yellow	*-*	*-*	[50]
18SA17	*Brevibacterium*	*antiquum*	+, orange	*-*	*-*	[51]
18FA7	*Caballeronia*	*udeis*	-	*-*	*-*	-
19SA8	*Comamonas*	*piscis*	+, yellow	*-*	*-*	[52]
19FR17	*Deinococcus*	*taklimakanensis*	-	*-*	Gamma UV tolerant	[53]
19SR23	*Dermacoccus*	*nishinomiyaensis*	-	*-*	*-*	-
19SA79	*Erwinia*	*billingiae*	-	*-*	*-*	-
19SA17	*Flavobacterium*	*frigidimaris*	+, yellow	+	*-*	[54]
19SA32	*Janthinobacterium*	*rivuli*	+, purple	*-*	*-*	[55]
19SA80	*Lysinimonas*	*soli*	-	*-*	*-*	-
19FR27	*Massilia*	*glaciei*	-	*-*	*-*	-
19SR22	*Massilia*	*brevitalea*	+, yellow	*-*	*-*	[56]
19FR22	*Massilia*	*eurypsychrophila*	-	+	*-*	[57]
19FR + 1	*Massilia*	*psychrophila*	+, yellow	+	*-*	[58]
19SR39	*Massilia*	*niabensis*	+, yellow	*-*	*-*	[59]
18SA14	*Microbacterium*	*maritypicum*	+, yellow	*-*	*-*	[60]
19FR32	*Micrococcus*	*aloeverae*	+, yellow	*-*	*-*	[61]
19FR42	*Micrococcus*	*flavus*	+, yellow	*-*	*-*	[62]
19FR13	*Moraxella*	*osloensis*	+, purple	*-*	*-*	[63]
19FR43	*Noviherbaspirilum*	*psychrotolerans*	-	+	*-*	[64]
19SR16	*Paenarthrobacter*	*nicotinovorans*	-	*-*	*-*	-
19SR10	*Paenarthrobacter*	*nitroguajacolicus*	-	*-*	*-*	-
19SR6	*Paenibacillus*	*liaoningensis*	-	*-*	Endospore-forming	[65]
19SR5	*Paenibacillus*	*algorifonticola*	-	*-*	*-*	-
19SA40	*Paracoccus*	*aquimaris*	-	*-*	*-*	-
18FA17	*Paracoccus*	*Sanguinis **	-	*-*	*-*	-
18FA24	*Paracoccus*	*Yeei **	-	*-*	*-*	-
19FR5	*Paracoccus*	*marinus*	+, orange	*-*	*-*	[47]
19SR31	*Pedobacter*	*ginsenosidimutans*	-	*-*	*-*	-
18SA31	*Pedobacter*	*nototheniae*	+, pink	*-*	*-*	[66]
19SA47	*Pedobacter*	*ginsengiterrae*	+, pink	*-*	*-*	[67]
18SA30	*Pedobacter*	*roseus*	+, pink	*-*	*-*	[68]
19SR20	*Pseudarthrobacter*	*sulfonivorans*	-	*-*	*-*	-
19SR33	*Pseudarthrobacter*	*siccitolerans*	+, beige	*-*	xeroprotectant	[69]
19SA35	*Pseudomonas*	*baetica*	-	*-*	*-*	-
19SA54	*Pseudomonas*	*guineae*	-	+	*-*	[70]
18SA13	*Pseudomonas*	*graminis*	-	*-*	*-*	-
18SA5	*Pseudomonas*	*marginalis*	-	*-*	*-*	-
19SA55	*Pseudomonas*	*plecoglossicida*	-	*-*	*-*	-
19SA52	*Pseudomonas*	*proteolytica*	-	+	*-*	[71]
19SA49	*Pseudomonas*	*trivialis*	-	*-*	*-*	-
18FA37	*Pseudomonas*	*caspiana*	-	*-*	*-*	-
19SA57	*Pseudomonas*	*helmanticensis*	+, beige	*-*	*-*	[72]
18SA9	*Pseudomonas*	*turukhanskensis*	+, beige	*-*	*-*	[73]
19SA5	*Pseudomonas*	*zhaodongensis*	+, orange	*-*	*-*	[74]
19SA71	*Pseudomonas*	*cedrina*	+, yellow	*-*	*-*	[75]
19SA12	*Pseudomonas*	*xanthomarina*	+, yellow	*-*	*-*	[76]
19SA2	*Psychrobacter*	*namhaensis*	-	*-*	*-*	-
18FA12	*Rhodococcus*	*kyotonensis*	+, orange	*-*	*-*	[77]
18FA15	*Saraclodium*	*kiliense*	-	*-*	resistant to anti-fungal	[78]
19SA65	*Serratia*	*proteamaculans*	-	*-*	*-*	-
19SA78	*Shewanella*	*hafniensis*	-	*-*	*-*	-
18SA19	*Sphingomonas*	*glacialis*	+, yellow	+	*-*	[43]
19FR + 6	*Sphingomonas*	*psychrolutea*	+, orange	+	*-*	[79]
18SA34F	*Sporobolomyces*	*blumeae*	+, orange	*-*	freeze–thaw tolerant	[80]
19FR11	*Staphylococcus*	*Epidermidis **	-	*-*	*-*	-
19FR30	*Staphylococcus*	*Warneri **	-	*-*	*-*	-
19SA19	*Stenotrophomonas*	*rhizophila*	-	*-*	Osmoprotective substances production and antifungal activity	[81]
18SA18	*Stenotrophomonas*	*chelatiphaga*	-	*-*	*-*	-
19FR2	*Streptococcus*	*Mitis **	-	*-*	*-*	-
19FR67	*Streptococcus*	*Sanguinis **	-	*-*	*-*	-
18FA10	*Streptomyces*	*candidus*	-	*-*	Produce antibiotics	[82]
19SA39	*Variovorax*	*boronicumulans*	-	*-*	*-*	-

## Data Availability

Partial 16S and 18S rRNA gene sequences obtained in this study have been deposited in the GenBank databases under the respective accession numbers OP765705 to OP765887 and OP765274 to OP765276.

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
