# Peer review of "Comparison of Atmospheric and Lithospheric Culturable Bacterial Communities from Two Dissimilar Active Volcanic Sites, Surtsey Island and Fimmvörðuháls Mountain in Iceland"

_microorganisms, 2023, doi:10.3390/microorganisms11030665_

Round 1

Reviewer 1 Report

The study is clear, well written and interesting. The methods are sound and the caveats are discussed when necessary. It's a nice paper that will add to the (still limited) understanding of airborne communities and aerobiology literature. 

Author Response

Thank you for reviewing our paper and for your comments.

Reviewer 2 Report

The paper is really interesting and the topic of innovative concern. The Introduction is adequately structured, but materials and methods and results require some minimal checks. Indeed, the paper should report more details on the isolated strains, at least on the main representetives, by adding the taxonomic identificatio obtained by BLAST, the Accession Number of the sequence deposition on GenBank and the information of the closest relatives. After this, they can refer to each isolate with the taxonomic ID obtained in the study for a better comprehension and valorization of paper. This could be done also by implementing the Supplementary Table 1, by adding the Accession Number of closest relatives and the similarity percentage.

Some suggestions, as follow:

Line 243. The methods for phisiological survival properties need to be better described.

Table 2.A Table showing the number of isolates obtained from each location and sample could be helpful.

Did the authors submit the sequence on public database, as GenBank? A supplementary table with all identified isolates should be also added, together with the Accession Number assigned by Gen Bank and their next relatives.

Table 2. In this Table the authors should report only the name of the strain and the taxonomic identification obtained and adequately described in the previous section, as suggested before. 

The paper deserves publication after minor revisions.
